# Ultra-Processed Foods in University Students: Implementing Nutri-Score to Make Healthy Choices

**DOI:** 10.3390/healthcare10060984

**Published:** 2022-05-25

**Authors:** Joan-Francesc Fondevila-Gascón, Gaspar Berbel-Giménez, Eduard Vidal-Portés, Katherine Hurtado-Galarza

**Affiliations:** 1Facultat de Comunicació i Relacions Internacionals Blanquerna, Blanquerna-Universitat Ramon Llull, 08001 Barcelona, Spain; 2Economy Department, EU Mediterrani-Universitat de Girona, 08015 Barcelona, Spain; gaspar.berbel@mediterrani.com (G.B.-G.); katherinehurtado@live.com (K.H.-G.); 3Advertising, Public Relations and Marketing Department, Faculty of Communication and International Relations, Blanquerna-Universitat Ramon Llull, 08001 Barcelona, Spain; eduardvp0@blanquerna.url.edu

**Keywords:** healthy eating, university students, ultra-processed food, feeding habits, Nutri-Score, nutrition labeling

## Abstract

Going to university means that many students assume, for the first time, responsibilities associated with living outside the family group, such as controlling eating habits. A survey was conducted among 161 university students in order to find out their perceptions regarding their type of diet, their knowledge of nutrition issues, their consumption of ultra-processed foods, and to evaluate the Nutri-Score labeling system as an aid in choosing healthier products. This is a cross-sectional observational study. Participants who have the perception of following a healthy diet show a more adequate BMI, regularly practice sports and read labels (nutritional information, expiration date, and ingredients). In general, the university students surveyed do not read the labels (64%) and find it difficult to identify the degree of wholesomeness of what they consume. Furthermore, they buy products based on the external information on the packaging (72%). The consumption of ultra-processed products, such as alcohol and soft drinks, is higher in those who live outside their family homes. The Nutri-Score labeling system is perceived as an aid for better product choice (89%).

## 1. Introduction

University students face new demands in their adult life, assuming new responsibilities and challenges typical of this stage, such as living outside the family residence and having to buy and cook their own food [1]. In relation to food, they do not always select healthy options—diets of low nutritional quality, far from the Mediterranean diet (frequently considered very natural or healthy), are frequent [2].

Students endure long hours of physical and mental effort, and, in many cases, go to the study center having eaten incomplete or very light meals [3], which can contribute to lower academic performance [4]. During the university stage, habits are consolidated, and in most cases, last over time. Therefore, we considered university students as a population sensitive to health promotion and prevention activities, since their lifestyles have a direct impact on their physical, psychological and mental development [5].

College settings are often places with a high availability of refined foods high in simple sugars and fats; where there is a greater restructuring of eating habits; where the use of ICT and social networks favor the consolidation of sedentary activities; with great ignorance regarding eating habits. This context favors the proliferation of restrictive and unbalanced diets, which increase the risk of developing eating behavior disorders, more frequently seen in women [6].

Carlos Ríos, the creator of the Realfooding movement, points out that there are many problems with having an unhealthy diet. It is dangerous living in a society in which we do not eat real food, and even more so when the brands sold to us in supermarkets are predominantly unhealthy ultra-processed foods. Within an environment controlled by the food industry, which keeps the population continually buying its products against their health [7], real food proposes a lifestyle that seeks to banish ultra-processed foods. The real food movement is a way of trying to educate people about a healthy diet and fight against processed food, which is considered negative. Real food refers to minimally processed foods whose industrial processing has not worsened the quality of the composition or negatively interfered with its naturally present healthy properties. Within real food, we can find minimally processed foods, whole foods, and unprocessed raw materials, and good processed foods or real foods with beneficial, harmless industrial or artisanal processing with respect to their health properties.

Ultra-processed foods are the opposite of real food; edible industrial preparations are made from different synthetic substances or products, with different processing techniques. They usually have five or more ingredients, and among them are added sugars, refined flours, refined vegetable oils, additives, and salt. Industrially processed ingredients achieve more durable, attractive, tasty, and highly profitable business, thus, they are sold and consumed above the rest of the options. Ultra-processed foods are usually dense in calories per unit of weight, poor in nutrients, widely available throughout our environment, and are promoted with strong marketing and advertising campaigns. They are marketed through large multinationals, covering almost 80% of the groceries sold in supermarkets. All this explains why ultra-processed foods are more successful than unprocessed or minimally processed foods [8]. This classification, according to Realfooding, is necessary to facilitate healthy food choices (real food) over unhealthy ones (ultra-processed), leaving behind the obsolete classification of many dietitians that is based solely on the chemical composition of the food, which spreads on social networks [9].

The classification of the Realfooding movement is based on a system developed a decade ago by researchers at the School of Public Health at the University of São Paulo. The system, called NOVA, qualifies foods according to their degree of processing [10].

Ultra-processed foods predominate in the food systems of the most industrialized countries [11]. Their consumption is indicated as a risk factor by the increase in obesity, measured by BMI, both in adolescents and adults [12], and is also associated with body fat in childhood and adolescence [13]. In addition, several studies show the relationship between ultra-processed foods and the risk of cardiovascular, coronary, cerebrovascular diseases, and cancer [14,15]. A recent study, carried out on 139 healthy adolescents between the ages of 13 and 19, revealed that a higher intake of ultra-processed foods was associated with greater oxidative DNA damage [16]. Other studies show the existence of a direct relationship between the consumption of ultra-processed foods and increased mortality [17,18,19].

The prevalence of obesity has increased in recent decades. In the first major health survey, conducted in Spain (ENS, 1987), obesity prevalence was 7.7% within the adult group, and in 2001 it doubled; in 2006, it reached 15.6%; in 2012 it stood at 15.5%; in 2014 at 16.9%; in 2017 at 17.4%. In 30 years, it has multiplied by 2.3. Men show greater obesity and being overweight than women [20].

Jaume Marrugat, head of the group of the Center for Biomedical Research in the Cardiovascular Diseases Network at the Hospital del Mar Institute, speaks of a pandemic of being overweight and obese, which has exploded since the year 2000 with a gradual and massive irruption of carbohydrates, sugary drinks, industrial pastries, and the abandonment of customs close to the Mediterranean lifestyle [21].

A possible cause of being overweight and obese is the imbalance between energy intake and expenditure. Thus, physical activity, together with good eating habits, are the main factors. Several observations in some countries showed that humans can consume recommended daily calories (i.e., not in excess) and become overweight. Obesity does not fully obey a linear and quantitative relationship between ingested and spent calories. Frequent physical activity—more than 2 times per week—is associated with a reduction in excess weight [20].

The Ministry of Spain, at the end of 2018, announced the adoption of the Nutri-Score nutritional front label, created by the French Public Health Agency, with the aim of helping consumers to judge the nutritional quality of packaged foods [22].

Frontal packaging labeling (EFE) systems characterize the nutritional quality of food, can reduce the time and cognitive effort in processing the labels, and identify the healthiest food options [23]. There are two major systems in front labeling:

(a) Nutrient-focused systems. These provide information on certain critical nutrients for health, whose excess intake increases the risk of obesity or other non-communicable diseases. In general, they report on kilocalories, fats, saturated fats, trans fats, salt/sodium, and/or sugars. Examples of this system would be nutritional traffic light labeling and warning labeling, among others.

(b) Summary systems. These are where information arises from an algorithm, based on a global evaluation of the product, in relation to its nutritional content, by synthesizing its nutritional quality in a symbol, icon, or score. In this second group, the Nutri-Score label would be present.

In this context, the Nutri-Score is a way to communicate to the consumers directly related to the real food movement. However, it has been recently shown that around 57% of industrial foods with Nutri-Score A/B are ultra-processed foods [24], and there is a study that blames industrial foods for childhood obesity [25,26]. Australian studies of the health star rating system [27] showed how the HSR is simple, uncluttered, easy to understand, and useful for a quick comparison across products. The nutrition information is positive, however, there is a perceived lack of transparency in the criteria used to determine the number of stars. Thus, numerous well-rated industrial foods by Nutri-Score are absolutely not “healthy” at all. Nutri-Score summarizes, therefore, a set of unintelligible numbers and terms of nutritional values, located on the back of the packaging, based on a colored logo that is easily understood by consumers. It informs consumers about the nutritional quality of food, facilitating the ability to identify and contrast its nutritional quality, and the choice of options with better nutritional quality [28]. The adoption of Nutri-Score by public health authorities is justified by studies that show the authenticity and effectiveness of its algorithm, as well as its superiority compared to other existing front-end labeling [28]. It is debatable, however, that there is a scientific reason for justifying the adoption of this score. The only scientific justification would be that the adoption of such scores worldwide would be really associated with decreased risks of chronic disease prevalence. A study of Spanish consumers indicated that Nutri-Score was well perceived and the characterization of nutritional quality was easy to understand [29], which allows consumers the possibility of making a better choice of products and verifying the nutritional levels [30].

In practice, Nutri-Score allows consumers, at a glance, to compare the nutritional quality at the time of purchase:-Foods belonging to different “families” (categories): for example, comparing the quality of yogurts with dessert creams; or foods consumed for breakfast, such as cereals with alternatives, such as sliced bread, cookies, biscuits, industrial pastries, etc.-Food of the same category: for example, cereals, such as mueslis versus chocolate cereals; fruit cookies versus chocolate cookies; different types of pizza with each other; types of drinks (water, fruit juice or soft drinks).-The same type of food, but different brands: for example, canned sardines of different brands.

The Nutri-Score nutritional logo (also known as 5C for its five colors—see Figure 1) details five types of nutritional quality, going from green (associated with the letter A) to red (associated with the letter E), based on the calculation of an algorithm according to scientifically validated public health criteria [23]. The algorithm underlying Nutri-Score is a system that was developed in Great Britain in 2005 by a team of Oxford researchers to regulate advertising aimed at children and was validated by the UK Food Standards Agency (FSA) [31,32]. In 2015, the Higher Council of Public Health of France [33] was in charge of delimiting a frontal labeling adaptation of the Anglo-Saxon algorithm that led to the FSAm/HCSP score (initials of the agencies of both countries).

This system is based on the attribution of points based on the nutritional composition per 100 g or 100 mL of a product (Figure 2).

On the one hand, a calculation is based on the content of nutrients considered “unfavourable” or critical from a nutritional point of view, to which a score of 0 to 10 points is attributed according to their content in kilocalories, simple sugars, saturated fatty acids and sodium (points A). On the other hand, “favorable” nutrients or ingredients (proteins, dietary fiber, and the percentage of fruits, vegetables, legumes, oleaginous fruits, olives, walnuts, and rapeseed oils) are considered and are assigned a score from 0 to 5 points (C points). The first sum of points corresponding to nutrients, points A, is made, and based on these, the total points of C are subtracted, without taking proteins into account.

The final score falls within the range of −15 to +40. Based on four predefined limits, the result is classified into five categories according to nutritional quality, and is represented in five colored circles, ranging from dark green to red, from the best to the worst nutritional quality. The largest circle, oversized, indicates the overall nutritional quality of the product. The association of colors and letters (A/B/C/D/E) seeks to guarantee greater understanding. All the elements used to calculate the Nutri-Score (which has received some critics) appear in the table of the mandatory nutritional declaration and in the list of ingredients located on the back of the packages, which guarantees transparency in the calculation and in the possibility of verifying the representation of the color attributed by Nutri-Score.

In the present study, we analyzed the perception of a group of university students, as a particularly sensitive group from the point of view of nutritional habits [34,35] with a low interest in reading nutritional labeling [30], who submit to information sources that prioritize instantaneity [36] a natural effect in the context of the broadband society and cloud journalism [37]. We surveyed their knowledge of nutrition, their degree of understanding of the Nutri-Score nutritional front labeling, their perceptions about food, and their level of knowledge of nutrition issues. An indirect effect of the study and the survey method used was to inform the participants of nutritional aspects and the function of the Nutri-Score label.

This observational cross-sectional study focuses on analyzing eating habits and nutritional knowledge in a sample of university students, as well as the Nutri-Score labeling system. We considered this sample as attending to the problems of leaving home, isolation, and new relationships, and the stress produced by a stronger level of university studies.

## 2. Materials and Methods

This was an observational study, where data was collected cross-sectionally through an online questionnaire. The elements collected refer to the perception of the diet of the participants, university students, their anthropometric data, their health habits, their knowledge of nutritional aspects, their consumption of ultra-processed foods, and the knowledge and acceptance of the labeling system, Nutri-Score. The survey was conducted in the 2020–2021 season in Barcelona.

### 2.1. Demographic Data

The sampling is intentional, non-probabilistic and of the snowball type [38], using social and contact networks of the Mediterrani University School, Barcelona (attached to the University of Girona). The final filtered sample was 161 students, consisting of 96 women and 65 men. The mean age is 21.8 years (SD = 3.8 years), 18 years old (3.1%), from 19 to 21 being the majority (68.3%), 22 years old (14.9%), and 23 years or older (13.7%).

The inclusion criteria in the study were being a university student at a university in the Spanish territory and knowing the Spanish language. A total of 68.3% of those surveyed studied at a public university compared to 31.7% who studied at a private university. A total of 26.7% of the students belonged to courses in social and legal sciences, 26.1% in arts and humanities, 20.5% in health sciences, 13.7% in engineering and architecture, and 13.0% in science.

Regarding the type of residence during the course, 44.1% resided in the family home, 35.4% in a shared flat, 18.0% live alone, and 2.5% live in university residences. A total of 12.4% have some pathology related to food (diabetes, hypertension, allergies, gastric problems, and others) that requires a special diet.

The research questions are:-Do university students have a wrong perception about what a healthy diet is?-Do they know and differentiate unprocessed from processed and ultra-processed products?-Will Nutri-Score be a measure and a new ally for a better and easy understanding of the front (nutritional) labeling of packaged products?-Will Nutri-Score labeling help make better decisions when choosing healthy products?-The objectives of the study are:-Collect indicators on eating habits in a group of university students, qualitative and quantitative (rates of consumption of ultra-processed foods).-Collect indicators on the perception of their diets and nutritional knowledge.-Measure the degree of knowledge and acceptance of the Nutri-Score label.

### 2.2. Questionnaire

Within the survey methodology, an online questionnaire of closed questions was carried out on Google Forms. The types of questions were structured in the following blocks:Sociodemographic (residence, sex, and date of birth).Typology of studies and university.Weight and height.Perceptions about food.Differentiation of ultra-processed foods.Frequency of consumption of ultra-processed foods and beverages.Reading and understanding nutritional labeling.

The BMI (Body Mass Index) was calculated from the anthropometric data of weight and height, and was classified according to the criteria of the Spanish Society for the Study of Obesity (SEEDO) into 6 categories: underweight (BMI < 18.5), normal weight (18.5 ≤ BMI < 24.9), grade I overweight (24.9 ≤ BMI < 26.9), grade II overweight (27 ≤ BMI < 29.9), type I obesity (29.9 ≤ BMI < 34.9) and type II obesity (35 ≤ BMI < 39).

#### 2.2.1. Frequency of Consumption and Knowledge in Nutrition

For the frequency of consumption of ultra-processed foods, the participants were asked to indicate the weekly rate of their consumption of soft drinks, sweet and salty snacks, industrial bakery, cereal and energy bars, energy drinks, processed meat products, soups and instant noodles, non-alcoholic beer and wine, whisky, gin, rum, and vodka.

For knowledge of nutrition, they were asked about the recommended daily amounts of salt and sugar consumption according to the WHO.

To read and understand food labels, participants were asked to classify a list of 10 products into three possible categories: (1) processed or minimally processed, (2) processed, or (3) ultra-processed.

#### 2.2.2. Notoriety of Nutri-Score System

Regarding the notoriety of the Nutri-Score system, the students were shown images of products with the Nutri-Score seal to assess the level of recognition of the logo, the interpretation, and their level of understanding compared to other types of front labeling.

Regarding statistical analyses and methods, data analysis was performed with the statistical package IBM-SPSS (v. 22). We carried out a descriptive analysis and different association tests between variables in order to answer the research questions raised.

The association tests performed were the Chi-square (association between categories), *t*-test, and ANOVA (association between categories of two or more levels and metrics). For comparisons between more than two post hoc means, the Bonferroni test was used.

## 3. Results

A total of 181 cases participated in the study; once the data had been refined, 161 remained definitive. The reasons for exclusion were not completing the survey or giving non-logical information, such as non-logical dates of birth.

### 3.1. Demographic Data

Among the participants in our sample (see Table 1), 85.7% had normal weight, 3.1% were underweight, and 11.2% were overweight.

### 3.2. Questionnaire

#### 3.2.1. Frequency of Consumption and Knowledge in Nutrition

Regarding the perception that university students have about their own diet, 42.3% considered it to be quite healthy or very healthy, 45% considered it normal, and only 11.8% affirmed that they have an unhealthy or unhealthy diet.

There was an association between the BMI and perception of their diet (F = 4.42; *p* = 0.014). The BMI was higher in those who perceived their diet as unhealthy compared to those who considered it normal or healthy. The contrasts (Bonferroni type) show significant differences between the normal and healthy or very healthy groups: the normal group shows between 0.2 and 2.4 points less BMI (CI95%) than the group that considered their diet as healthy or very healthy. (t = 1.30; *p* = 0.019).

Regarding the practice of physical activity, 16% stated that they never did it (19% of men compared to 15% of women), and 62% stated that they did it two or more times a week (65% of men compared to 60% of women). There was an association between the practice of physical activity and the perception of the diet: those who practiced some physical activity two or more times a week perceived their diet as healthier compared to those who never practiced any activity or practiced it more sporadically (*X*^2^ = 9.85, *p* = 0.043).

As shown in Table 2, only 25.5% knew exactly what the recommended daily intake of salt and sugar was by the World Health Organization (5 g of salt and 25 g of sugar).

There was an association between age and knowledge of the amounts of salt and sugar consumption recommended by the WHO. The youngest tended to be more confused or unaware of the recommended daily amounts of salt and sugar (t = 4.59; *p* < 0.0001). Those who were most correct in these amounts were the oldest: 29.2% in the 22-year-old group were correct.

A total of 94% of overweight participants, in any of their degrees, said they did not know the recommended amounts of sugar and salt, compared to 70% of those who were normal weight or low weight (*X*^2^ = 5.05; *p* = 0.025).

A total of 70.8% of university students stated that they knew the difference between unprocessed or minimally processed, processed, and ultra-processed products, according to the Nova classification (see Table 3).

A total of 74.0% of the women declared knowing the differences between defendants, compared to 66.2% of the men, however, there is no association between sex and said differentiation (*X*^2^ = 1.142; *p* = 0.285).

Regarding the knowledge and correct classification of products according to whether they are unprocessed, minimally processed, processed, or ultra-processed, the degree of success varied according to the product and was not associated with gender (see Table 4).

A total of 32.5% of the participants correctly classified Serrano ham as a processed product, according to the Nova classification. As for canned lentils, 57% classified them as unprocessed or minimally processed, 40% did it correctly (processed), and 62% knew to which group the wine belongs (processed). As for nuts and eggs, almost all made a correct classification. A total of 65.8% were wrong when classifying oat flakes as a processed product since, in the NOVA system, it was classified as non-processed or minimally processed food. A total of 65.8% of those surveyed were correct in the classification of crab sticks (ultra-processed). Only 30.7% of the participants were correct with Maria biscuits as an ultra-processed food. A total of 85.1% correctly classified the Red Bull/Monster drinks in the ultra-processed group, although they were not so clear in the case of Nocilla/Nutella spreads since 50.9% classified them incorrectly as if they were processed foods when they are actually ultra-processed.

A total of 36% of the participants declared that they always read the nutritional information on food labels, 54% did sometimes, and 10% never do so. Among those who said they “Never Read” the nutritional labeling of foods (Table 5), 31.3% declared “Excess information” displayed as the reason. Within the “Other response category”, such a small font size predominates.

The type of studies where it is most stated to “Always Read the labels” to know the nutritional composition is that of health sciences (60.6%). There is an association between “Type of Studies” and “Reading” (*X*^2^ = 14.19; *p* = 0.007). Among all the participants who declared they “Never” or “Only Sometimes” read nutritional labeling, those who belong to the branches of sciences and social and legal sciences stood out (86% and 72%, respectively).

A strong association was observed between “Diet Perception” not healthy, normal, or healthy and “Label Reading” (*X*^2^ = 24.9; *p* < 0.0001). A total of 65.4% of the participants who said they follow a healthy diet affirmed they always read the labels, compared to 18% of those who said they follow an unhealthy diet.

Within the product information, nutritional information is the most consulted element with 49.1%, followed by the expiration date or preferential consumption and ingredients, with 24.8% and 13.7%, respectively. Both men and women showed interest in the nutritional information on food labels.

As can be seen in Table 6, most of the participants found it neither easy nor difficult to identify healthy products, although 24.8% found it quite or very difficult to do so. Having difficulty or not identifying the level of health is not associated, in our sample, with either sex (*X*^2^ = 7.8; *p* = 0.05) or with weight status (*X*^2^ = 3.7; *p* = 0.293).

A total of 72% of the participants admitted to having bought a food product only for the information shown on the front of the package. The effect of the front information on the purchase does not vary with sex (*X*^2^ = 1.3; *p* = 0.257).

High rates of consumption—once a week or more—of ultra-processed foods (the 10 indicated in the study questionnaire, see Table 7), are more frequent in men, and significantly in the products: energy drinks, alcohol, and cereal bars.

The type of university—public or private—does not affect the majority of the consumption rates of ultra-processed products. Only in soft drinks (*X*^2^ = 4.6; *p* = 0.032) and breakfast cereals (*X*^2^ = 5.2; *p* = 0.023) is the consumption somewhat higher in the participants of the public system; in instant noodles and spas in those of the private one (*X*^2^ = 4.9; *p* = 0.027).

The type of residence does not affect the majority of the consumption rates of ultra-processed products. The higher consumption occurs only in soft drinks (*X*^2^ = 4.1; *p* = 0.043) and alcohol (*X*^2^ = 5.1; *p* = 0.024), where the participants are outside the family residence while studying.

#### 3.2.2. Notoriety of Nutri-Score System

The label reading frequency is not associated, in our sample, with the manifesting of any pathology that requires a special diet—diabetes, hypertension, allergies, intolerances, gastric problems… (*X*^2^ = 1.510; *p* = 0.470).

In the questionnaire, the participants were shown a food product with the Nutri-Score image, highlighting category E on the front of the package. A total of 65.8% of those surveyed claimed to recognize or had seen the Nutri-Score system when buying products. After showing them the said image, they were asked to mark the meaning of the Nutri-Score label. As Table 8 shows, the most frequent response (77.0%) refers to the fact that it reports on the degree of nutritional quality, the correct option.

To find out the degree of help that the Nutri-Score system represents, the respondents were shown two products from the same category: one with the Nutri-Score logo, and the other with the so-called “Guideline Daily Amounts” (CDO) nutritional label on the front. A total of 88.8% chose the Nutri-Score scheme as the easiest label to understand.

The online questionnaire showed, in its final part, an explanation of the Nutri-Score system, subsequently asking the participants to what extent the information provided by the said system would help them to implement healthier habits. Most of the participants (89%) indicated that the Nutri-Score label would help them quite a lot to introduce healthy foods into their diet, more so than the women’s group, with 94% compared to 82% of men (*X*^2^ = 6.9; *p* = 0.032).

Despite the fact that the majority correctly understands the meaning of the Nutri-Score label (nutritional quality grade), it is higher in the women group—women reached 87% compared to the global 77% (*X*^2^ = 11.9; *p* = 0.003).

## 4. Discussion and Conclusions

We observed that 87.5% of the students participating in the study have a BMI within the normal weight range. A total of 11.2% of students are overweight or obese; this is soothing to note due to the relationship between the consumption of ultra-processed foods and mortality [17,18,19].

It is striking, however, that most of those who are overweight or obese perceive their diet in a distorted way, qualifying it as normal or even very healthy. Neither sex nor age, the type of university, studies, or branches of knowledge influences how the study participants perceived their diet. A total of 86.9% of the participants (university students) considered their diet as normal, very healthy, or quite healthy. Only 11.8% admitted to having an unhealthy diet—the rest did not know. However, the results showed that this perception is wrong. Despite the fact that university students (70.8%) claim to know how to differentiate the foods that are within the NOVA food classification system, the data collected shows just the opposite.

The consumption of ultra-processed foods by the respondents is too high to consider that they have a normal or healthy diet. Weekly, 71% admitted to consuming soft drinks, 83% for snacks, 83% for nuggets, 67% for pastries, and 68% for alcohol. A total of 25.5% did not know the adequate or optimal amount of the daily consumption for salt and sugar. More information, however, is considered to be positive, granting people a better choice of products, and checking nutritional levels [30].

A total of 20.2% acknowledge not knowing how to differentiate between unprocessed, processed, or ultra-processed products. This was confirmed in the product classification test, where 50% misclassified Nocilla/Nutella, 70% for oat flakes, 71% for canned lentils, and 68% for ham. Our data demonstrate the great lack of knowledge that exists about what is eaten.

An alarming fact, which reflects the lack of information and awareness about what is eaten, is that 64% declared that they sometimes or never read the nutritional information of the products. A total of 33% did not know what the Nutri-Score label indicates, although some studies indicated that Nutri-Score is well perceived and the characterization of nutritional quality is easy to understand [28].

Those who perceive their diet as healthier are participants with a lower BMI, exercise regularly and read product labels—especially nutritional information, expiration dates, and ingredients.

Women show a better knowledge of processed products, in addition to the fact that they tend to consume less ultra-processed products than men, especially alcohol, bars, and energy drinks. Therefore, some did seem to manage their new responsibilities and greater autonomy more effectively, without the quality of their diet being conditioned by their new circumstances (more sedentary life and availability of less healthy products), thus, ceasing to be a vulnerable group from a nutritional point of view [3].

Being overweight, in the participants, is associated with no physical activity and a lack of nutritional information—94% of those who are overweight say they did not know the recommended daily amounts of salt and sugar.

Of the mandatory food information that is reflected in the labeling [30], the students’ focus was mainly on nutritional information (49.1%), being the most consulted element by almost half of the students, and was higher in the case of women. However, there is a greater interest in the expiration date or preferential consumption than in the ingredients, the latter being essential for the identification of healthy products.

The participants, in general, do not read the labels—their nutritional information. A total of 64% only sometimes or never read the labels, mainly due to the excess of information, not considering it important, and the difficulty of understanding the symbols. A total of 33% did not know the meaning of the Nutri-Score label. It was difficult for them to know the degree of the healthiness of the products, to identify the healthiest, and 25% considered that it was quite or very difficult to determine if they were healthy. They recognized that labels such as the Nutri-Score would help them to better choose the products they consume. Those who live outside the family home consume more alcohol and soft drinks.

A total of 72% of students admit to having bought food products due to the influence of the information displayed on the front of the package. Men seem somewhat more easily influenced (77% compared to 69% for women).

A total of 65.8% of the people surveyed admit having seen the Nutri-Score nutritional label on the front, being the highest level of recognition in the case of women. The majority (49.1%) correctly affirmed that its meaning is the degree of nutritional quality of a product. When comparing the labels of two products, one with the Nutri-Score scheme and the other with the CDO label, it was observed that 88.8% of the participating university students say they better interpreted the product labeled with Nutri-Score; therefore, it seems to be an easy-to-use system for understanding [23]. Likewise, for 88.8% of those surveyed, this label would help them to introduce more healthy foods into their diet, more so in the group of women. Therefore, due to the easy interpretation of the Nutri-Score logo and the lack of other tools that help participants make healthier decisions, Nutri-Score is postulated as a good guidance system for healthier food purchase decisions.

Previous research studies on the understanding of the Nutri-Score labeling, or other nutritional labels that help people make healthy food choices in the university population, are scarce.

The sample is somewhat small (161 individuals) and the non-probabilistic sampling prevents making generalizations from the data collected. Therefore, we are dealing with an exploratory study. A larger, probabilistic sample would allow broader and more robust generalizations to be made.

Some relevant information could not be verified, such as the self-reported data on weight, height, or the perception of food, thus, certain variables may contain bias or inaccuracy to a certain degree.

For subsequent lines of research, it is recommended to carry out a new survey on a larger, probabilistic sample of university students about the Nutri-Score logo, and its influence on the purchase decision. In addition to this, exploring and analyzing new trends and habits, such as the influence and consumption of certain channels and influencers, as well as some apps to analyze the food barcode in situ, is recommended. Currently, there are few brands that include the Nutri-Score label, as it is not required. However, it is being adopted by more products every day, and awareness of proper nutrition is increasing.

## Figures and Tables

**Figure 1 healthcare-10-00984-f001:**
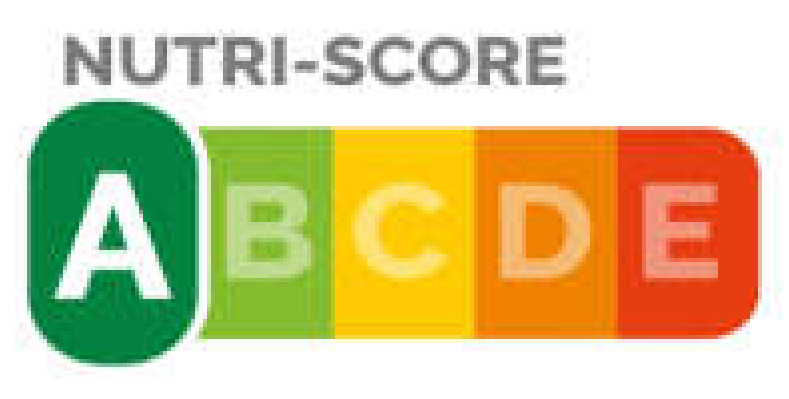
Nutri-Score logo.

**Figure 2 healthcare-10-00984-f002:**
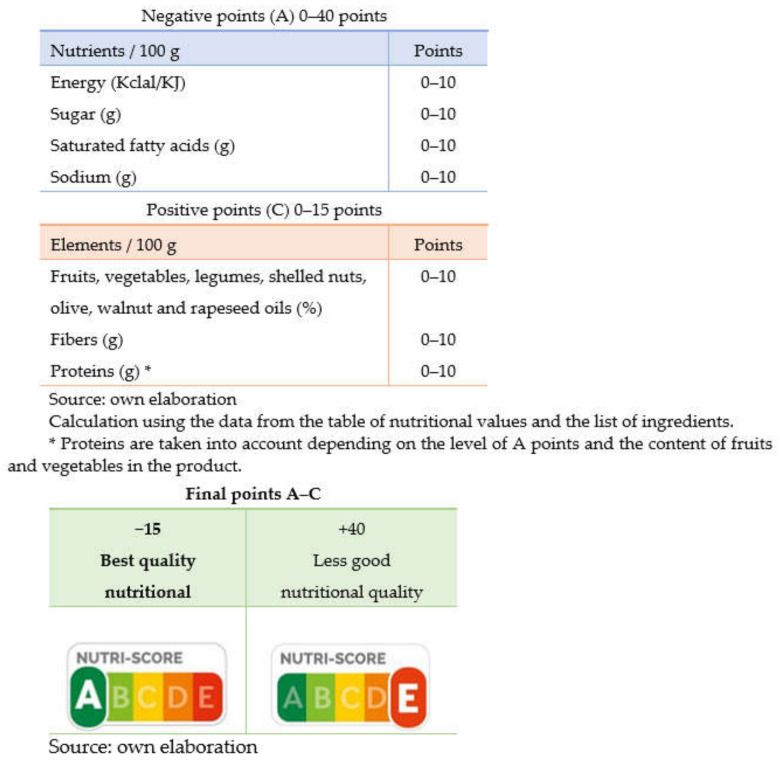
Nutri-Score algorithm calculation.

**Table 1 healthcare-10-00984-t001:** Distribution of university students according to the BMI classification criteria proposed by the Spanish Society for the Study of Obesity (SEEDO).

	Man		Woman		Total	
	*n*	%	*n*	%	*n*	%
Underweight	1	1.5	4	4.2	5	3.1
Normoweight	55	84.6	83	86.5	138	85.7
Underweight degree I	6	9.2	2	2.1	8	5.0
Underweight degree II	2	3.1	6	6.3	8	5.0
Obesity kind I	1	1.5	0	0.0	1	0.6
Obesity kind II	0	0.0	1	1.0	1	0.6
Total	65	100.0	96	100.0	161	100.0

Source: own elaboration.

**Table 2 healthcare-10-00984-t002:** Knowledge about the daily consumption of salt and sugar recommended by the WHO.

	*n*	%
10 g of salt and 20 g of sugar	45	28.0
5 g of salt and 15 g of sugar	34	21.1
5 g of salt and 25 g of sugar	41	25.5
I don’t know	41	25.5
Total	161	100.0

Source: own elaboration.

**Table 3 healthcare-10-00984-t003:** Knowing the difference between unprocessed or minimally processed, processed and ultra-processed products.

	Total		Man		Woman	
	*n*	%	*n*	%	*n*	%
Sí	114	70.8	43	66.2	71	74.0
No	47	29.2	22	33.8	25	26.0
Total	161	100.0	65	100.0	96	100.0

Source: own elaboration.

**Table 4 healthcare-10-00984-t004:** Hits in the classification of products by sex.

	% of Hits in Classify Products		Chi Squared	
	Man	Woman	*X* ^2^	*p*
Ham	28	35	2.1	0.355
Pot lentils	42	38	0.393	0.822
Wine	65	60	1.19	0.551
Walnuts	100	96	1.86	0.172
Eggs	98	94	0.70	0.403
Oatmeal	28	37	1.63	0.442
Surimi	67	65	0.084	0.772
Maria cookies	72	68	0.253	0.615
Red Bull/Monster	88	83	0.587	0.444
Nocilla/Nutella	51	48	0.12	0.735

% of correct answers classifying products as unprocessed or minimally, processed or ultra-processed. Within men and women. *p* = degree of significance of the Chi-square test. Processed in bold, unprocessed or minimally processed in normal font, ultra-processed in gray. Source: own elaboration.

**Table 5 healthcare-10-00984-t005:** Reasons for not reading food product labels.

	*n*	%
Information overload	5	31.3
I do not consider the information they provide important	4	25.0
They use symbols and texts that are difficult to understand	4	25.0
Other	3	18.8
Total	16	100.0

Source: own elaboration.

**Table 6 healthcare-10-00984-t006:** Identification of healthy products.

	Total		Man		Woman	
	*n*	%	*n*	%	*n*	%
Quite/Very difficult to identify	40	24.8	14	21.5	26	27.1
Neither easy nor difficult to identify	83	51.6	34	52.3	49	51.0
Quite/Very easy to identify	28	17.4	9	13.8	19	19.8
I don’t know	10	6.2	8	12.3	2	2,1
Total	161	100.0	65	100.0	96	100.0

Source: own elaboration.

**Table 7 healthcare-10-00984-t007:** Consumption of ultra-processed foods by gender.

	High Consumption Rate Ultra-Processed (%)		Chi-Squared	
	Man	Woman	*X* ^2^	*p*
Soft drinks	79	65	3.6	0.059
Snacks	83	83	0.002	0.966
Industrial bakery	69	55	3.2	0.074
Cookies	71	65	0.67	0.412
Breakfast cereals	79	77	0.042	0.837
Cereal and energy bars	65	46	5.5	0.019 **
Energy drinks	39	18	8.7	0.003 **
Nuggets, sausages…	80	78	0.082	0.775
Instant noodles and soups	46	40	0.686	0.408
Alcohol	79	57	7.72	0.005 **

% of high consumption (weekly or more) compared to zero or occasional consumption. Within men and women. *p* = degree of significance of the Chi-square test. (**) Significant association at 1%. Source: own elaboration.

**Table 8 healthcare-10-00984-t008:** Interpretation of the Nutri-Score system.

	Total	
	*n*	*%*
Amount of sugar	6	3.7
Number of calories	9	5.6
Amount of fat	3	1.9
Amount of vitamins	2	1.2
Nutritional quality grade	124	77.0
Processing degree	14	8.7
I do not know	3	1.9
Total	161	100.0

Source: own elaboration.

## Data Availability

Not applicable.

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
