# Peer review of "Ultra-Processed Foods in University Students: Implementing Nutri-Score to Make Healthy Choices"

_healthcare, 2022, doi:10.3390/healthcare10060984_

Round 1

Reviewer 1 Report

  1. Line 2 on the title, use a colon (:) instead of a full stop after the word students. This is because the first sentence explains the first one thus a colon should be joining the two not a full stop
  2. Line 36, be consistent in your referencing style. Remove the full stop before Sánchez-Ojeda and De Luna-Bertos, 2015
  3. A sentence 'Industrially processed ingredients to achieve more durable, attractive, tasty and highly profitable products so that they are sold and consumed above the rest of the options.' on line 57-59 is not complete. Please re-write to make sense
  4. Line 61 add the word 'and' before the word 'promoted'
  5. Line 79 'revealed' instead of 'reveals'
  6. Line 80, 'was' instead of 'is' as you are reporting the results of a study already conducted
  7. Line 95 should read 'are the main factors' instead of 'is the main factor' because you mentioned two factors not one
  8. General comments: Please make a good use of the word 'and' throughout your document
  9. The introduction section is unnecessarily very long. Consider reducing it to at least two pages with only important highlights.
  10. Line 204 should read 'system: Nutriscore'. Please not the colon and not comma
  11. Lines 212, 213 and 217, do not start a sentence with a number and check throughout the document
  12. Line 218, do not use...... if you mean and others please use etcetera or etc 
  13. Line 221, 'carried out' is a repetition delete it
  14. Line 235 to 238 why do you use ; instead of comma (,) after each aspect? Use comma instead
  15. In table 1 and 2, why do you write numbers with commas to mean decimals?? Example 85,7 to mean 85.7? Re-write correctly with decimals not commas for the relevant numbers in this table.
  16. Results general comment: Should be written in the past tense because it is reporting a study already done. At the moment the author is using the present tense, this should be corrected throughout the manuscript.
  17.  Revise the use of full stops in the middle of sentences

Author Response

Please see the attachment (thanks for the comments)!

Reviewer 2 Report

Dear authors,

I found your research very interesting. It covers how efficient the communication between health authorities (through Nutriscore) and the consumers and their knowledge about the nutritional value of food.

As general recommendations to improve your manuscript, I advise you to work on the following:

I would modify the introduction section to be in the following order: 1) Why are you observing that sample (the problems of leaving home, and so on), 2) What is the problem of having an unhealthy diet (problems related to it), 3) The real food movement as a way of trying to educate people for a healthy diet (here you explain why is processed food considered "bad", according to the paragraph above), 4) the Nutriscore as a way to communicate to the consumers (maybe you can relate it with the real food movement). 

The Results section should be joined with the discussion, as in the discussion you repeat a lot of information from the results section. Also, I think you can explore the results based on more information, crossing with what you have in the introduction. 

I also recommend you revise your text to improve your written English. English is not my native language, but I can see some sentences lack clarity, and in some cases, you used Spanish words (please correct that!).

I pointed out other details in the attached file while I read your manuscript.

I hope my comments can help you improve your work!

Author Response

(The authors gave the same response as above.)

Reviewer 3 Report

A - General:

This observational cross-sectional study aimed at studying the eating habits of university students, an at-risk population for their dietary habits, more specifically to collect indicators on their eating habits, their perception of their diets and nutritional knowledge, and the degree of knowledge and acceptance of the Nutri-score label.

Main results showed that:

  • “participants who have the perception of following a healthy diet show a more adequate BMI, regularly practice sports, and read labels (nutritional information, expiration date, and ingredients).”
  • “the university students surveyed do not read the labels (64%) and find it difficult to identify the degree of wholesomeness of what they consume, they buy products based on the external information on the packaging (72%).”
  • “the consumption of ultra-processed products such as alcohol and soft drinks is higher in those who live outside their family homes.”

The study has been well carried out with appropriate statistical tests for such a study. The results are interesting, although, as mentioned by authors in the Discussion, a broader student sample would be probably necessary in the future. However, there is a “good” heterogeneity of the student population as regards with age, sex, university type, the nature of the teachings of the university, type of residence, and BMI.

My main concerns are as follows:

  • The Introduction is somewhat too long: notably the section about Nutri-score is too long because this has been already published several times. Some references, with perhaps a short section, would be sufficient.
  • It is several times suggested that Nutri-score would be associated with healthier food choices (lines 9, 104, 124, 446…): this is wrong. For example, it has been recently shown that around 57% of industrial foods Nutri-score A/B are ultra-processed foods (Ebner et al., How are the processing and nutrient dimensions of foods interconnected? An issue of hierarchy based on three different food scores. International Journal of Food Science and Nutrition 2022.). There is also the study by Richonnet et al. about industrial foods for children (Richonnet et al., Nutritional Quality and Degree of Processing of Children’s Foods Assessment on the French Market. Nutrients 2022;14:171). See also Australian studies with the Health Star Rating System. Since UPF are associated with higher risks of chronic diseases and early mortality, this cannot be written. Thus, numerous well rated industrial foods by Nutri-score are absolutely not “healthy” as such.
  • Otehrwise, it lacks a more generic analysis of the interconnection between NOVA classification and the Nutri-score, which leads to the erroneous conclusion lines 447-449: “Therefore, due to the easy interpretation of the Nutri-score logo and the lack of other tools that help participants make healthier decisions, Nutri-score is postulated as a good guidance system for healthier food purchase decisions.”. This is not supported by science. This is not because Nutri-score appears easy to interpret and because there is a lack of other tools (which is not really true, again: there is other tools) that you can conclude that “Nutri-score is postulated as a good guidance system for healthier food purchase decisions”. This appears more as a point of view than scientifically based.
  • Are there studies about UPF consumption in university students? If, yes, you should mention them in the Introduction.
  • At the end of manuscript there is no section about:
    • Author Contributions
    • Funding
    • Institutional Review Board Statement
    • Informed Consent Statement
    • Potential links/conflicts of Interest

More generally, we sometimes have the implicit impression, at some places of the manuscript, that the authors are trying to “sell” the Nutri-score without being really critical and by omitting certain studies... It's somewhat a little embarrassing.

B - Specific:

Lines 21-23: should be at the end of the Introduction.

Lines 54-66: it lacks references.

Lines 94-95: “The main cause of overweight and obesity is the imbalance between energy intake and expenditure”: that's not really accurate. Several observations in some countries showed that you can consume recommended daily calorie (i.e., not in excess) and become overweight. Obesity does not fully obey to a linear and quantitative relationship between ingested and spent calories: this is more complex than that. Calorie quality also matters. However, the next part of the sentence is OK lines 95-96.

Lines 119-122: “The adoption of Nutri-score by public health authorities is justified by studies that show the authenticity and effectiveness of its algorithm, as well as its superiority compared to other existing front-end labelling” (Chantal and Hercberg, 2017): This is not a scientific reason for justifying the adoption of this score. The only scientific justification would be that the adoption of such scores worldwide would be really associated with decreased risks of chronic disease prevalence. Is that really the case? In addition, what did you mean by “authenticity and effectiveness of its algorithm”? It is unclear, scientifically speaking.

Lines 186-193: Research questions should be in the Method section.

Line 387: there are no real conclusions in this section? Please, amend/elaborate.

Author Response

Please see the attachment. Thanks for the comments!

Round 2

Reviewer 2 Report

Dear authors,

I'm sorry for being "the bad reviewer". I think the introduction is almost perfect, with a few corrections to make, however, the materials and methods and the results and discussion can be improved in terms of structure.

I advise you to structure the materials and methods as:

2.1 Demographic data

2.1 Questionnaire

2.1.2 Part x

2.1.3 Part Y

2.... Statistical analysis

The results and discussion should come as one and follow the same structure as the methodology:

Results and discussion:

3.1 Demographic data

3.1 Questionnaire

3.1.2 Part x

It was possible to observe that xxx; the author yyyy, et al, (2022) found the same results zzzz...

3.1.3 Part Y

The results are shown in Table 15, and are in accordance with XYZ

The conclusion section should be short and show the main conclusions of your study.

I hope it helps, please don't hate me :)

Author Response

Dear reviewer.

Don't worry about your comments! They are very useful and they help to improve the final result of the article. We comment it:

"I'm sorry for being "the bad reviewer"." You are a good reviewer!

"I think the introduction is almost perfect, with a few corrections to make, however, the materials and methods and the results and discussion can be improved in terms of structure.

I advise you to structure the materials and methods as:

2.1 Demographic data

2.1 Questionnaire

2.1.2 Part x

2.1.3 Part Y

2.... Statistical analysis

The results and discussion should come as one and follow the same structure as the methodology:

Results and discussion:

3.1 Demographic data

3.1 Questionnaire

3.1.2 Part x

It was possible to observe that xxx; the author yyyy, et al, (2022) found the same results zzzz...

3.1.3 Part Y

The results are shown in Table 15, and are in accordance with XYZ"

We have adapt the article to your nice proposal.

"The conclusion section should be short and show the main conclusions of your study."

Certainly, we have reduced parts of the conclusion section.

"I hope it helps, please don't hate me :)" We don't hate you; we are grateful with you. Thanks!